# LEARNED HARDWARE/SOFTWARE CO-DESIGN OF NEURAL ACCELERATORS

## ABSTRACT

The use of deep learning has grown at an exponential rate, giving rise to numerous specialized hardware and software systems for deep learning. Because the design space of deep learning software stacks and hardware accelerators is diverse and vast, prior work considers software optimizations separately from hardware architectures, effectively reducing the search space. Unfortunately, this bifurcated approach means that many profitable design points are never explored. This paper instead casts the problem as hardware/software co-design, with the goal of automatically identifying desirable points in the joint design space. The key to our solution is a new constrained Bayesian optimization framework that avoids invalid solutions by exploiting the highly constrained features of this design space, which are semi-continuous/semi-discrete. We evaluate our optimization framework by applying it to a variety of neural models, improving the energy-delay product by 18% (ResNet) and 40% (DQN) over hand-tuned state-of-the-art systems, as well as demonstrating strong results on other neural network architectures, such as MLPs and Transformers.

## 1 INTRODUCTION

The compute requirements of deep learning are growing at a double exponential rate (Hernandez & Brown, 2020), with more powerful models requiring exponentially more compute to train. This growth has been enabled by large systems of hardware accelerators, like GPUs and TPUs (NVIDIA, 2017; Jouppi et al., 2017). However, the continued scaling of these systems is limited by issues of power density, cooling, and memory, so we need to improve computational efficiency.

Efficiency improvements can be sought at each layer of the deep learning stack, from better learning algorithms (Kingma & Ba, 2014), to improved neural network architectures (Tan & Le, 2019), to deep learning compilers (Chen et al., 2018), to specialized DNN accelerators that increase hardware efficiency (Chen et al., 2014a; 2016). In this paper, we focus on the low-level software and hardware portions of this stack, with the goal of automatically optimizing the $energy \times delay$ product of executing a particular model on a hardware accelerator. We consider two components from the deep learning stack: the hardware accelerator and the software compiler that maps a model onto that hardware. This area is commonly referred to as hardware/software co-design, and since it requires human expertise from multiple disciplines (software engineers, compiler writers, hardware architects), it is typically driven by manual heuristics or heuristic-based search (Yang et al., 2020b).

We propose a different approach, recognizing that for a given DNN model, this hardware/software co-design can be framed as a joint search of the space of all of the valid mappings and hardware architectures that can correctly execute the model. We formally parameterize this space based on prior work (Parashar et al., 2019), and we find that standard optimization techniques, including off-the-shelf Bayesian optimization, perform poorly because the design space is semi-discrete and the vast majority of the points in the space are infeasible. Prior work (Nardi et al., 2019) makes a similar observation, noting that (1) complex constraints such as hardware area and energy budget limit the feasible parameter values (i.e. small feasibility set), (2) some constraints are unknown until after a sample point has been evaluated (i.e. unknown feasibility).

Our solution casts the search as a bilevel optimization problem, as shown in Figure 1. The outer loop optimizes over hardware architectures, while the inner loop optimizes over software mappings for a given architecture. Both of these are heavily constrained black-box global optimization problems

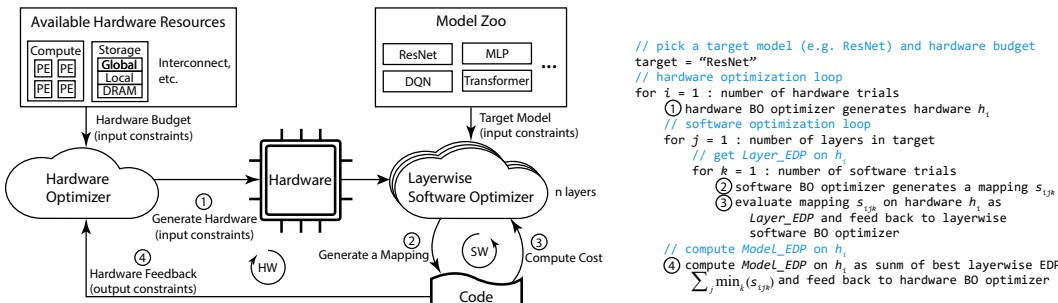

Figure 1: Overview of BO-based nested search for hardware/software co-design.

that require expensive simulations to obtain performance estimates. We therefore propose a nested, constrained Bayesian optimization (BO) formulation that uses Bayesian models of hardware and software performance to guide the search towards promising regions of the design space. Our approach is extensible to a variety of different neural network architectures, and we make the code publicly available.

We find that when compared against the state-of-the-art manually-designed hardware accelerators that use heuristic software mappings, our BO-based approach provides significant improvements in the speed and energy efficiency of the resulting system, improving the energy-delay product (EDP) by 16.0% to 40.2% on a series of neural networks. The key to our solution is our robust BO software optimizer, whose consistent efficiency allows our approach to scale to this huge search space.

This paper makes the following contributions:

- We present the first system that automatically co-optimizes both the hardware architecture and software mapping phases of DNN accelerator design using a principled and systematic search algorithm.

- We present a constrained formulation of hardware and software design for BO, a challenging problem given the high ratio (90%) of invalid hardware and software designs.

- We present a nested hardware/software formulation of BO that is extensible to other hardware accelerator designs.

- We provide model-specific hardware and state-of-the-art results on multiple models.

## 2 A FORMAL REPRESENTATION OF SOFTWARE AND HARDWARE

Hardware/software co-design typically performed manually, but we believe that this vast design space is best navigated by an intelligent search process. To facilitate this automation, this section formally defines the hardware and software design spaces.

### 2.1 PARAMETERIZING THE DESIGN SPACE

*Software design points* can be parameterized by the loop ordering, loop tiling, and computational parallelism of the seven-level loop nest used to compute a convolutional layer (see appendix), as has been noted by recent work (Parashar et al., 2019; Yang et al., 2020b). These software parameters are subject to hardware constraints, such as the quantity and layout of processing elements (PEs) and the size of storage elements.

*Hardware parameters* can be broken down into a two broad categories:

*Resource configurations* represent the physical aspects of hardware, such as buffer sizes, tile sizes, and the cluster size of global buffers, as well as the layouts of the PE array and of the global buffer.

*Dataflow configurations* represent the usage of the PE array that are implemented in hardware, such as the blocking factors and degree of parallelism at the PE level, which also determines the communication patterns among PEs.

Figure 2 shows two possible design points for a 1D convolution. Both design points tile and parallelize the channel (C) dimension. To the right of each component in the architecture is a set of loops that specifies the control logic for the component, which can be broken down into temporal streaming (`for` loops) and spatial distribution (`parallel_for` loops). For example, in the architecture on the left, the global buffer distributes across the PEs 1 weight from 4 separate channels (`c2`), and the

PEs perform all operations that the weight participates in. In this design, all data reuse is captured within the PE, so the global buffer need not store anything. By contrast, the architecture on the right distributes a single output element across the PEs to compute partial sums, which are stored in the global buffer across iterations. Both these design points consist of the same architectural components, but the dataflows vary, imposing different constraints on the software. The appendix shows the details of the parameterization of a more practical 2D convolution.

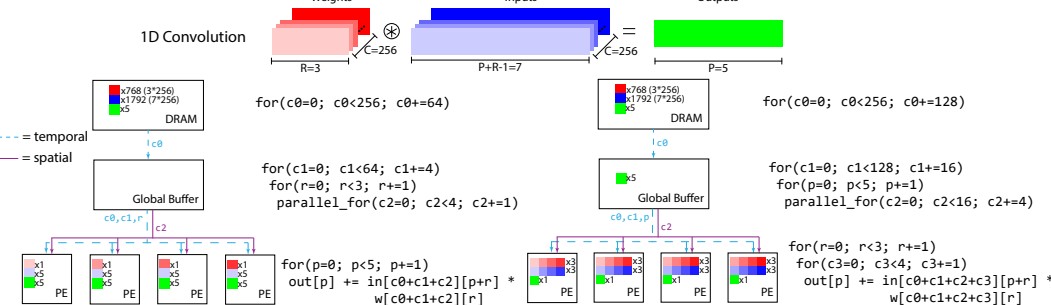

Figure 2: Two architectures computing a 1D convolution.

## 2.2 CONSTRAINTS IN THE DESIGN SPACE

There are several reasons why the vast space of hardware and software parameters is filled with impractical or invalid design points. First, hardware designs are fundamentally constrained by area (the total amount of compute and storage resources) and factors such as available memory bandwidth. Second, the design cost and latency of additional area grow super-linearly (Shao et al., 2019), which leads to many impractical design points.

Software constraints are generally governed by feasibility instead of practicality and predominantly depend on the hardware configuration and the specific neural network workload. For a specific hardware accelerator, there is a limited number of available resources, so the software optimization problem can be viewed as a search for the most efficient use of hardware PEs and buffers. For example, the loop blocking optimization factors a neural network across multiple hardware storage buffers—and the feasible factorizations are constrained by the size of the hardware buffers.

## 3 BAYESIAN OPTIMIZATION

### 3.1 OVERVIEW

Bayesian optimization (Jones et al., 1998; Brochu et al., 2010; Shahriari et al., 2015) is an effective approach for the optimization of expensive, possibly noisy black-box functions. BO has been used to optimize hyperparameters (Snoek et al., 2012), configure algorithms (Hutter et al., 2011), optimize A/B experiments (Letham et al., 2019), and more. For our problem, we have a parameterized representation and access to a simulator. Since one of our main concerns is sample efficiency, Bayesian optimization is particularly suitable.

The actual cost of evaluation depends on the experimental infrastructure, but in general, it is much more expensive to evaluate a hardware design choice than to evaluate software optimizations, because hardware design can take hours (to produce a hardware simulator or an FPGA) to days or even months (to produce an ASIC).

Bayesian optimization has two major components: (1) a surrogate model provides a Bayesian posterior probability distribution that predicts potential values of the objective function. (2) an acquisition function uses the model to identify the next point to evaluate.

### 3.2 GAUSSIAN PROCESSES

A common surrogate model is a Gaussian process (GP) (Rasmussen & Williams, 2006) due to its simplicity and flexibility. A GP is prior distribution over the space of functions comprised of a mean function $m(\mathbf{x})$ and a covariance, or kernel function $k(\mathbf{x}, \mathbf{x}')$. Suppose we are given a dataset of $N$ input/output pairs over a bounded domain $\Omega$ with $D$ input dimensions and scalar outputs. For brevity, we write this as $(X, \mathbf{y})$, where $X \in \Omega^{N \times D}$ and $\mathbf{y} \in \mathbb{R}^N$. The posterior predictive distribution over function values $f$ for a new input $\mathbf{x}$ is given by $P(f \mid \mathbf{x}, X, \mathbf{y}) = \mathcal{N}(\mu(\mathbf{x}), \sigma^2(\mathbf{x}))$, where

$$\mu(\mathbf{x}) = K_{\mathbf{x}X} K_{XX}^{-1} (\mathbf{y} - \mathbf{m}_X) + m(\mathbf{x}),$$
$$\sigma^2(\mathbf{x}) = k(\mathbf{x}, \mathbf{x}) - K_{\mathbf{x}X} K_{XX}^{-1} K_{\mathbf{x}X}^{\top}.$$

Where $K_{XX}$ is a matrix formed by evaluating the kernel on $X$, $K_{\mathbf{x}X}$ is the vector of kernel evaluations between $\mathbf{x}$ and $X$, and $\mathbf{m}_X$ is the vector of mean function evaluations on the input dataset.

A common choice for the kernel is squared exponential. Given two input vectors $\mathbf{x}_i$ and $\mathbf{x}_j$, this is defined as $k(\mathbf{x}_i, \mathbf{x}_j) = \alpha^2 \exp\left(-\frac{\|\mathbf{x}_i - \mathbf{x}_j\|^2}{\ell^2}\right)$. $\alpha$ and $\ell$ are kernel hyperparameters.

Another kernel that we find particularly useful is a linear kernel on top of explicit features. Given a feature mapping $\phi(\mathbf{x}) : \mathbb{R}^D \to \mathbb{R}^K$, the linear kernel can be written as $k(\mathbf{x}_i, \mathbf{x}_j) = \phi(\mathbf{x}_i)^\top \phi(\mathbf{x}_j)$. When we have strong prior information about the relevant feature interactions that govern the black-box function, this kernel allows us to encode these interactions directly and produces a more sample-efficient posterior estimate.

In cases where the observations from the black-box function are noisy, we can add a noise kernel $K_{\text{noise}} = \tau^2 \mathrm{I}$ to $K_{XX}$, where $\tau^2$ is a hyperparameter. This implies a Gaussian observation likelihood.

Following common practice, we use the constant mean $m(\mathbf{x}) = c \quad \forall\, \mathbf{x}$. All kernel and mean hyperparameters are learned by maximizing the marginal likelihood of the GP on the current dataset.

### 3.3 ACQUISITION FUNCTIONS

A critical component in the BO framework is the choice of acquisition function $a(\cdot)$ that assigns each design point a value that represents the utility of testing this point. Two commonly used acquisition functions are expected improvement (EI) and lower confidence bound (LCB).

EI computes the amount we expect to improve upon the current best observed objective value $y_* \equiv \max\{y_i\}_{i=1}^N$ by evaluating a design point $\mathbf{x}$. Formally, it can be written as

$$a_{\text{EI}}(\mathbf{x}) = \int_{-\infty}^{\infty} \max(y_* - f, 0) P(f \mid \mathbf{x}, X, \mathbf{y}) \mathrm{d}f.$$

where $f$ is the latent function from the surrogate model, and $y_*$ is the best value observed.

LCB (Srinivas et al., 2009) provides an explicit tradeoff between the predictive mean and variance and is defined as

$$a_{\text{LCB}}(\mathbf{x}) = \mu(\mathbf{x}) + \lambda\sigma(\mathbf{x}).$$

Where $\lambda$ represents a tradeoff parameter. A small $\lambda$ promotes greater exploitation, and a large $\lambda$ promotes greater exploration. We found $\lambda = 1$ to work well in our experiments. Beyond these, there are many other possible acquisition functions that could be used in future exploration (Thompson, 1933; Hennig & Schuler, 2012; Hernández-Lobato et al., 2014; Frazier, 2009).

### 3.4 CONSTRAINTS

In our problem, the vast majority of the design space will produce invalid solutions. When the constraints are a known function of the input features, we can directly account for them as input constraints. Otherwise, we must run the simulation and treat invalid points using an output constraint. Here, we will describe these constraint types, and how they are incorporated into BO.

*Input constraints* are explicit constraints that are used when optimizing the acquisition function. They directly prevent the search from suggesting points that will violate the constraints. As some constraints are non-linear, this optimization is itself very challenging, as it is a global optimization problem with non-convex constraints. In the unconstrained case, maximizing the acquisition function often takes a hybrid approach: generating a random initial set of points and refining them by gradient ascent. Maintaining feasibility with non-convex constraints is far more challenging, however.

We therefore optimize the acquisition function in a simple way by performing rejection sampling on the design space: we randomly sample parameters until we obtain 150 feasible points, and then choose the one the maximizes the acquisition function. On average the sampling takes 22K random samples to get a pool of 150 feasible points. We have found that practically this is a simple yet effective strategy for our problems, we leave more involved optimization schemes for future work.

*Output constraints* are used when we do not know the form of the constraint a-priori and must run the simulator to test feasibility. This is also referred to as an "unknown" constraint, and BO has been adapted to incorporate a constraint model in addition to the regression model (Gelbart et al., 2014). These simultaneously learn about the constraint boundaries while modeling the objective.

Let $\mathcal{C}(\mathbf{x})$ denote the event that $\mathbf{x}$ satisfies constraint $\mathcal{C}$. Constrained BO uses a Bayesian classifier to model $P(\mathcal{C}(\mathbf{x}))$. It is relatively straightforward to adapt a GP regressor to classification (Rasmussen & Williams, 2006).

Under a Bayesian classifier, the acquisition function $a(\mathbf{x})$ is modified to account for the probability that the constraint is satisfied, with 0 utility if it is not satisfied.

$$\bar{a}(\mathbf{x}) = \mathbb{E}[a(\mathbf{x})\mathrm{I}[\mathcal{C}(\mathbf{x})]] = P(\mathcal{C}(\mathbf{x}))a(\mathbf{x}).$$

Where $\mathrm{I}[\mathcal{C}(\mathbf{x})]$ is the indicator function that evaluates to 1 if the constraint is satisfied and 0 otherwise. We therefore maintain two models: one regression model to capture the objective and one classifier to model the constraint in order to avoid evaluations in infeasible regions.

## 4 BAYESIAN OPTIMIZATION FOR HARDWARE/SOFTWARE CO-DESIGN

### 4.1 OVERVIEW OF NESTED HARDWARE/SOFTWARE OPTIMIZATION

Provided the constraints discussed in Section 2 and the BO formulation from Section 3, we propose a nested approach for co-optimizing hardware/software parameters. The overall approach is outlined in Figure 1. The goal is to find the optimal hardware parameters for a neural model and the optimal set of software parameters for each layer in the neural model. Since software constraints depend on a feasible hardware design, we first propose the hardware parameters, then for that hardware co-optimize the software mapping.

Specifically, let $\mathbf{x}_h$ and $\mathbf{x}_s$ denote the set of hardware and software parameters in the parameter space to be optimized. In the nested search process, we first use the hardware optimizer to generate a design of hardware. In particular, we perform the hardware search in the space of possible hardware $\mathcal{S}_h$ to optimize all hardware parameters, where the objective is to minimize $f(\mathbf{x}_h \mid \mathrm{NN})$ which we define as the energy-delay product (EDP) of running the neural network (NN) model on the given hardware, assuming the optimal software mapping for each individual layer. This step produces a hardware specification and can be formalized as $\operatorname{argmin}_{h \in \mathcal{S}_h} f(\mathbf{x}_h \mid \mathrm{NN})$.

For the chosen hardware design, our framework performs the software search for each individual neural layer in its constrained software mapping space $\mathcal{S}_s \mid h, \mathrm{NN}_j$ to optimize the mapping parameters, where $\mathrm{NN}_j$ denotes the $j$th layer in the neural network model, and the objective becomes $f(\mathbf{x}_s \mid \mathbf{x}_h, \mathrm{NN}_j)$, which is defined as the EDP of running the layer $j$ on the fixed hardware. This step produces a design point that represents the best set of software mappings for each layers on the given hardware structure, and can be formalized as $\operatorname{argmin}_{s \in \mathcal{S}_s \mid h} f(\mathbf{x}_s \mid \mathbf{x}_h)$. The layerwise EDPs are then summed up as the EDP of the neural model, which is fed back to the hardware optimizer to generate the next hardware setting.

The iterative search between hardware and software will repeat for a user-defined number of trials. In this work, we set 50 for hardware search and 250 for software search. The combination of hardware and software that achieves the best EDP during the optimization process becomes the final model-specific hardware structure and layer-specific software mappings. A random sample is used in the first iteration of both the hardware and software search. In our Bayesian optimization (BO) framework, we use separate BO models to search in the hardware and software space. We now describe their design considerations, particularly the choice of kernel and feature transformation.

### 4.2 BO FOR OPTIMIZING HARDWARE ARCHITECTURES

**Kernel design.** The main design choice for BO is the GP kernel to use. For the hardware search, we choose a linear kernel on top of feature transformations that represent the relationship between the different parameters. This feature transformation allows us to explicitly encode domain knowledge. For example, by comparing the layout parameters of the 2D PE array and global buffer we can obtain the ratio between these adjacent storage layers, which correlates to the maximal degree of parallel buffer accesses in each dimension. The details of the features are given Figure 13 in the appendix. We also add a noise kernel to deal with noise in the hardware evaluation. This is because the software optimizer is not guaranteed to find the best software mapping for each layer. There is some randomness in the software search, and therefore independent runs of software optimization for a fixed hardware design may yield different results.

**Constraints.** There are both known and unknown constraints in the hardware search. The known constraints, such as the compute and storage budget, are treated as input constraints that reject invalid samples. The unknown constraints have to do with feasibility (if there exist valid software mappings

of neural layers onto the hardware, and if the valid mappings can be sampled during the software optimization). Following Section 3, these constraints are treated as output constraints and are modeled by a GP with a squared exponential kernel.

### 4.3 BO FOR OPTIMIZING SOFTWARE MAPPINGS

**Kernel design.** Similar to hardware optimization, we use a linear kernel and transform the parameters to features that encode relational information. As the hardware is fixed in the search of software mappings, we are able to compute features such as buffer usage, which potentially help make the predictions more accurate. The evaluation of a mapping on a given hardware is deterministic in our infrastructure, thus there is no need for a noise kernel in the GPs.

**Constraints.** As both the hardware and neural model are known during software optimization, all constraints are known and are treated as input constraints that automatically reject invalid samples.

## 5 EVALUATION

### 5.1 METHODOLOGY

**Infrastructure.** We conduct our evaluation on Timeloop (Parashar et al., 2019), which is an open-source infrastructure for evaluating the hardware design and software optimization of DNN accelerators. Timeloop represents the key architecture attributes of DNN accelerators that realize a broad space of hardware structure and topology, which generate an accurate projection of performance and energy efficiency for DNN workloads. In the evaluation, Timeloop takes two inputs: 1) the hardware configuration, which consists of the hardware-related parameters, and 2) the software mapping, which consists of the software parameters that describe the mapping. As most accelerators are designed for neural network inference, we limit the use case to inference in this work and leave training for future work.

**Workloads.** To show that our solution automatically produces efficient hardware for a variety of neural networks, we use our BO framework to optimize critical layers from CNNs (ResNet (He et al., 2016) and DQN (Mnih et al., 2013)), as well as an MLP and Transformer (Vaswani et al., 2017).

**Experimental Setup.** We use Eyeriss (Chen et al., 2016), a state-of-the-art DNN accelerator, as our baseline. All workloads are evaluated on the Eyeriss implementation with 168 PEs (Chen et al., 2016) except for the Transformer model, which runs on the larger version of Eyeriss with 256 PEs (Parashar et al., 2019). In the software mapping search, we use Eyeriss's hardware specifications and search for the best software mapping for each neural layer. In the hardware search, we perform the search under the same compute and storage resource constraints as Eyeriss for each neural model. [1]

**Metrics.** Hardware accelerators are designed to achieve both speed and energy efficiency, so we adopt the widely used energy-delay product (EDP) as the objective. Since EDP values can vary by an order of magnitude, we normalize by dividing by the best (minimal) EDP value and take the reciprocal for optimization curves. For the hardware/software co-design, we report the EDP improvements of each neural model, which is averaged across all layers (see Figure 11 and 12 in the appendix). For software mapping optimizations, we report the layer-wise EDP improvements.

**Baselines.** In hardware search, we compare against constrained random search that repeatedly takes the first random sample in the design space that satisfies the constraints. In software search, we use constrained random search, TVM with XGBoost and TreeGRU (Chen et al., 2018), and out-of-the-box BO that optimizes in a continuous parameter space and rounds to the nearest valid parameters.

**Representation of the Search Space.** Our representation uses the original values of the parameters (normalized by the downstream GPs), including high-level features such as buffer and compute usage. We have tried a few other approaches to defining distance in this space, but find that our current representation is the most effective one. For example, we applied log transformations to the tiling factor to transform the nonlinear constraints to linear, but the GP suffers greatly from the transformed space where the distance is skewed.

---

[1] This work focuses on model-specific hardware, but hardware specialization provides larger benefits at a finer granularity, i.e. if different layers can execute on customized hardware. We leave this for future work.

## 5.2 SOFTWARE MAPPING OPTIMIZATION

We show the results of software mapping optimization first, as the capability of finding a good mapping is the base of evaluating a hardware design. Figure 3 shows the improvements of BO over our constrained random search formulation. Our BO formulation outperforms random search, both variants of TVM as well as a standard BO formulation that optimizes discrete parameters using a relax-and-round approach.

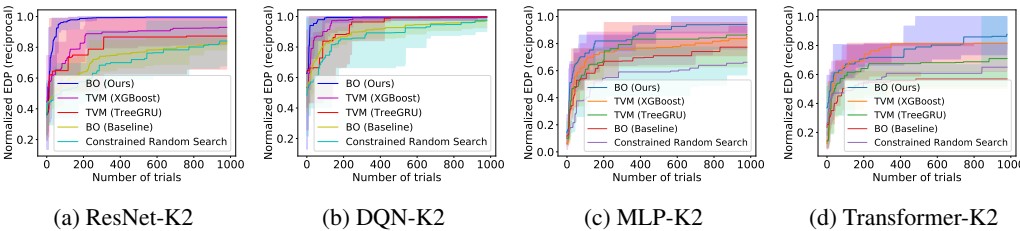

| (a) ResNet-K2 | (b) DQN-K2 | (c) MLP-K2 | (d) Transformer-K2 |

Figure 3: Software mapping optimization on layer 2 of ResNet, DQN, MLP, and Transformer. The y-axis shows the reciprocal of energy-delay product (EDP) (normalized against the best EDP value). Higher is better. Results for other layers can be found in the appendix. Best viewed in color.

## 5.3 HARDWARE CONFIGURATION OPTIMIZATION

Figure 4 shows the optimization curves for hardware/software co-design. The comparison of hardware search algorithms shows that BO provides consistently better performance than the constrained random search, and the comparison of software search algorithms shows the importance of mapping optimization in the co-design process. As shown in Figure 5a, we find that the designs searched by BO achieve significantly better EDP on all neural models compared to the state-of-the-art manually designed accelerator (18.3%, 40.2%, 21.8% and 16.0% for ResNet, DQN, MLP and Transformer respectively).

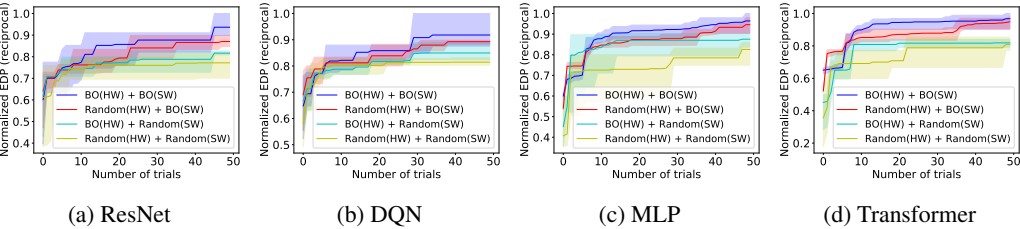

| (a) ResNet | (b) DQN | (c) MLP | (d) Transformer |

Figure 4: Hardware/software co-optimization. The x-axis shows the number of trials for hardware search, and 250 attempts are made to find the optimal software mapping for each layer in the model on the hardware specification. Best viewed in color.

## 5.4 ABLATION STUDIES

**Surrogate models and acquisition functions.** There exist popular variants for both the surrogate models and acquisition functions. In Figure 5b, we compare the surrogate models of Gaussian process (GP) with random forest (RF) and the acquisition functions of expected improvement (EI) and lower confidence bound (LCB). As shown, in the transformed feature space, GP generally performs better than RF, and LCB generally outperforms EI.

**Exploration vs. Exploitation.** The LCB acquisition function explicitly balances exploration and exploitation with a hyperparameter $\lambda$. To further study the impact of exploration vs. exploitation in the use of LCB, we test LCB with different $\lambda$ values in Figure 5c. We find that LCBs with $\lambda$ values that are greater than 0.5 provide stable performance in the optimization curves, while LCB with $\lambda = 0.1$ suffers from insufficient exploration.

## 5.5 ARCHITECTURAL INSIGHTS

To show that our automated design can produce new architectural insights, we provide a qualitative comparison of Eyeriss with our solution for DQN. Our design predominantly differs in the shape of the PE array, as well as in the number of memory buffers used. Eyeriss allocates the majority of its local buffer storage for filter weights, which are poorly utilized. Our design increases buffer utilization by storing multiple inputs and output elements.

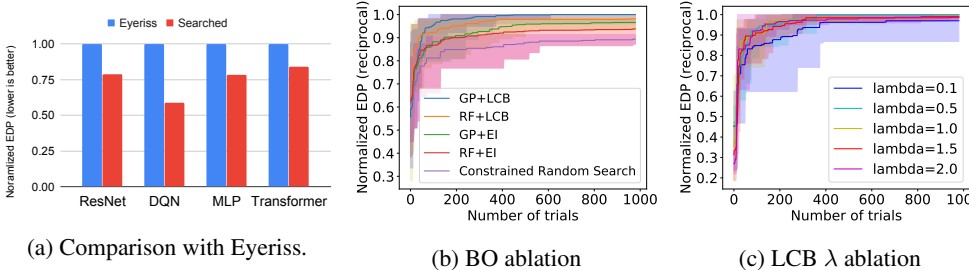

(a) Comparison with Eyeriss.    (b) BO ablation    (c) LCB $\lambda$ ablation

Figure 5: (a): Comparison between the SOTA accelerator (Eyeriss) and searched design. Results are EDPs normalized to Eyeriss, and lower is better. (b)-(c): Ablation studies on ResNet-K4. Higher results are better. (b) BO with different surrogate models and acquisition functions. (c) LCB acquisition function with different $\lambda$ values.

We can also plug our hardware configuration into the heuristic-based optimizer from prior work (Parashar et al., 2019) and attempt to find a software mapping. We do this using the $12 \times 14$ PE array from DQN. Timeloop's software optimizers are unable to find the same mapping that we do, with the best result being 52% worse than the baseline. This demonstrates the utility of a learned co-design approach that enables the software optimizer to be robust across different hardware architectures.

# 6 RELATED WORK

## 6.1 HARDWARE TO OPTIMIZE DNNs

Accelerators are specialized processors that provide significant performance and efficiency improvements by targeting specific workloads; they also typically require significant manual design. For deep learning, these primitives are often basic linear algebra subprograms (BLAS).

Prior work has designed specialized hardware to execute BLAS kernels. Google's TPU (Jouppi et al., 2017) uses large hardware structures called systolic arrays (Kung & Leiserson, 1979), and NVIDIA's GPUs have tensor cores (NVIDIA, 2017). DianNao (Chen et al., 2014a) computes fully connected layers of a neural network using multiply-accumulate trees, and its successor, DaDianNao (Chen et al., 2014b), improves data locality for large neural networks by tiling processing elements (PEs) around on-chip eDRAM banks. A more specialized version, ShiDianNao (Du et al., 2015), focuses on dense convolutional neural networks (CNNs), which exhibit regular computational behavior with high data reuse. Eyeriss (Chen et al., 2016) and Eyeriss v2 (Chen et al., 2019) also focus on CNNs, introducing a specific dataflow that exploits a reuse pattern exhibited by 2D convolutions. To improve performance scaling, Eyeriss v2 uses a more sophisticated interconnect than its predecessor, and it also introduces a variant that targets sparse CNNs. Prior work (Parashar et al., 2017; Zhang et al., 2016) has dealt with sparsity by suppressing zero-valued activations and storing and operating on compressed data. Many other domain specific architectures have been proposed to take advantage of local communication patterns (Farabet et al., 2011), 3D-stacked bandwidth memory (Kim et al., 2016; Gao et al., 2017), or multi-chip modules (Shao et al., 2019). Recent work (Yang et al., 2020b) presents heuristics for automatically synthesizing hardware using a domain-specific language.

## 6.2 SOFTWARE TO OPTIMIZE DNNs

Software compiler optimizations for reducing the compute and storage requirements of neural networks include loop blocking (tiling), loop reordering, and loop unrolling (Whaley & Dongarra, 1998; Mullapudi et al., 2016; Bondhugula et al., 2008). Compilers such as TVM (Chen et al., 2018) have used learned cost models to optimize execution efficiency. Similarly, Timeloop uses a grid or random search to optimize software mappings on a user-specified hardware architecture (Parashar et al., 2019). However, all previous software optimizers treat hardware as a black box and ignore interactions between hardware and software.

## 6.3 NEURAL ARCHITECTURE SEARCH

Neural architecture search (NAS) is a research area that seeks to automatically select the optimal neural network topology or architecture to select for each problem, often agnostic to the underlying hardware or software (Zoph & Le, 2016). NAS is disjoint to our work, as we seek to optimize lower-level hardware/software primitives, but an optimization approach that considers neural network architecture, software mappings, and hardware architecture could be future work.

### 6.4 BI-LEVEL OPTIMIZATION APPROACHES

Recent research has started to optimize pairs of these categories jointly. For example, EDD (Li et al., 2020) co-optimizes NAS and software compiler optimizations on embedded systems, but does not consider hardware architecture. Several prior papers have co-optimized NAS with hardware architecture (Jiang et al., 2020; Yang et al., 2020a; Lin et al.; Abdelfattah et al., 2020). These works either fix the software compiler optimizations (Jiang et al., 2020; Lin et al.; Abdelfattah et al., 2020) or tie the compiler optimizations to the hardware search (Yang et al., 2020a), which simplifies the search problem, but the compiler optimizations cannot be customized to different workloads once the hardware is synthesized.

Ours is the first work that systematically co-optimizes the space of both hardware and software compiler optimizations. This larger search space requires a principled search method, which motivates our constrained Bayesian optimization framework. Our nested search minimizes expensive hardware generation and allows us to optimize the hardware for the entire model, but still optimize software mappings for each layer of the neural network.

## 7 CONCLUSION

In this paper, we have cast hardware/software co-design as a Bayesian optimization problem. We have shown that standard mechanisms have difficulty navigating the complex, highly constrained design space, so we have presented a novel constrained formulation that allows the optimizer to efficiently identify desirable points in this design space. The use of machine learning to automate hardware/software co-design opens many opportunities for future work. For example, transfer learning could dramatically reduce design time across designs and models. The techniques described here are not limited to DNN architectures, which is significant because as we enter the golden age of computer architecture (Hennessy & Patterson, 2019), it is essential that we develop automatic mechanisms for architectural exploration that quickly produce custom hardware accelerators.

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

# A APPENDIX

## A.1 PARAMETERS AND CONSTRAINTS

| Type | Index | Hardware Parameters | Valid Range | Meaning |
|------|-------|---------------------|-------------|---------|
| PE | H1 | PE mesh-X | Factors of # PEs | Decide the arrangement of the 2-D PE array. |
| | H2 | PE mesh-Y | Factors of # PEs | |
| Local buffer | H3 | Input entries in Local buffer | 0 to # local buffer entries | Decide the partition of local buffer. The partition leads to sub-buffers with inflexible sizes. This is useful as the latency to access each smaller sub-buffer decreases. |
| | H4 | weights entries in Local buffer | 0 to # local buffer entries | |
| | H5 | outputs entries in Local buffer | 0 to # local buffer entries | |
| Global buffer | H6 | Global buffer instances | Factors of #PEs | Determine the arrangement of global buffer, and its connection between global buffer and per PE's local buffer (Local buffer of PEs along the X-axis shares the instances of global buffer along the X-axis). |
| | H7 | Global buffer mesh-X | Factors of PE-mesh-X | |
| | H8 | Global buffer mesh-Y | Factors of PE-mesh-Y | |
| | H9 | Global buffer block size | Factors of 16 | Determines the width of a global buffer entry |
| | H10 | Global buffer cluster size | Factors of 16 | Determines of the number of wider structures where multiple entries are ganged into |
| Dataflow | H11 | Dataflow option of filter width | 1, 2 | Options that determine the size of filter width in PE's local buffer |
| | H12 | Dataflow option of filter height | 1, 2 | Options that determine the size of filter height in PE's local buffer |

Figure 6: Hardware parameters.

| Type | Hardware Constraints |
|------|----------------------|
| PE | PE mesh-X (H1) * PE mesh-Y (H2) = # PEs |
| Local buffer | The sum of local sub-buffers (H3, H4, H5) does not exceed buffer size |
| Global buffer | Global buffer mesh-X (H7) * global buffer mesh-Y (H8) = # Global buffer instances (9) |
| Local buffer & global buffer (unknown) | A valid software mapping exists depending mainly on local buffer partition (H3, H4, H5) and global buffer arrangement (H6, H7, H8) |

Figure 7: Hardware constraints.

| Type | Index | Software Parameters | Valid Range | Meaning |
|------|-------|---------------------|-------------|---------|
| Loop blocking and degree of parallelism | S1 | Blocking factors of R | Factors of R | Determines the size (parallelism) of each type of data (inputs, weights and outputs) in each storage layer (except those that are in the hardware dataflow). |
| | S2 | Blocking factors of S | Factors of S | |
| | S3 | Blocking factors of P | Factors of P | |
| | S4 | Blocking factors of Q | Factors of Q | |
| | S5 | Blocking factors of C | Factors of C | |
| | S6 | Blocking factors of K | Factors of K | |
| Loop reorder | S7 | Loop order in local buffer | Permutations of non-1 factors | Affects the reuse of each type of data (inputs, weights and outputs) in each storage layer. |
| | S8 | Loop order in global buffer | Permutations of non-1 factors | |
| | S9 | Loop order in DRAM | Permutations of non-1 factors | |

Figure 8: Software parameters.

# B HYPERPARAMTERS FOR BO

In Figure 10 we report the hyperparamters for BO.

| Type | Software Constraints |
|---|---|
| Loop blocking and degree of parallelism | Product of all blocking factors of R (S1) equals R of the target neural layer |
| | Product of all blocking factors of S (S2) equals S of the target neural layer |
| | Product of all blocking factors of P (S3) equals P of the target neural layer |
| | Product of all blocking factors of Q (S4) equals Q of the target neural layer |
| | Product of all blocking factors of C (S5) equals C of the target neural layer |
| | Product of all blocking factors of K (S6) equals K of the target neural layer |
| Buffer capacity (local) | Inputs/weights/outputs sizes (S1-S6) cannot exceed corresponding local sub-buffer capacity |
| Buffer capacity (global) | Size of all types of data (S1-S6) does not exceed global buffer capacity |
| Parallelism | Product of blocking factors in global buffer X-axis (S1-S6) cannot exceed # PEs in X-axis |
| | Product of blocking factors in global buffer (S1-S6) cannot exceed total # PEs |

Figure 9: Software constraints.

| number of independent trials | 5 (HW), 10 (SW) |
|---|---|
| number of random data points | 50 (HW), 150 (SW) |
| number of warmup data points | 5 (HW), 30 (SW) |
| number of samples for EI | 1000 |
| lambda for LCB | 1.0 |

Figure 10: Hyperparamters for BO.

## C  NEURAL MODEL SPECIFICATIONS.

In Figure 11 and Figure 12 we report the specifications of neural models benchmarked in this paper.

## D  PARAMTERIZATION OF 2D CONVOLUTION

Listing 14 gives the seven-level nested loop that comprises a 2D convolution.

Figure 17 shows a design point for the CONV4 layer of ResNet. The architecture components are again the same as in the 1D example, but since the memory footprint is significantly larger, the PE can no longer capture all data reuse, so the Global Buffer must store large portions of the inputs and outputs.

## E  EXAMPLE PARAMETER VECTOR

Below are example vectors of hardware and software parameters our BO optimizes.

## F  ADDITIONAL RESULTS

### F.1  SOFTWARE OPTIMIZATION

In Figure 18 we show more examples of the software optimization over multiple layers of the different architectures. Our Bayesian optimization formulation consistently outperforms the baselines (Chen et al., 2018).

### F.2  ABLATIONS

In Figure 19 we compare different surrogate models and acquisition functions for Bayesian optimization of the software mapping. We found Gaussian processes with LCB to consistently outperform other alternatives.

| Model | Layers | Specifications |
|---|---|---|
| ResNet | ResNet-K1 | Filter size: 3×3
Output size: 56×56
# input channel: 64
# output channel: 64
Stride: 2 |
| | ResNet-K2 | Filter size: 3×3
Output size: 28×28
# input channel: 128
# output channel: 128
Stride: 1 |
| | ResNet-K3 | Filter size: 3×3
Output size: 14×14
# input channel: 256
# output channel: 256
Stride: 1 |
| | ResNet-K4 | Filter size: 3×3
Output size: 7×7
# input channel: 512
# output channel: 512
Stride: 1 |
| DQN | DQN-K1 | Filter size: 8×8
Output size: 20×20
# input channel: 4
# output channel: 16
Stride: 4 |
| | DQN-K2 | Filter size: 4×4
Output size: 9×9
# input channel: 16
# output channel: 32
Stride: 2 |

Figure 11: Specifications of ResNet (ResNet-18) (He et al., 2016) and DQN (Mnih et al., 2013)

In Figure 20 we investigate the robustness of LCB for software optimization using different values of $\lambda$. We found that $\lambda = 0.1$ tends to be too greedy, but that above $\lambda = 0.5$, LCB tends to be fairly robust.

| Model | Layers | Specifications |
|---|---|---|
| MLP | MLP-K1 | $d_{in}$: 512
$d_{out}$: 512 |
| | MLP-K2 | $d_{in}$: 64
$d_{out}$: 1024 |
| Transformer | Transformer-K1 | $d_{model}$ = 512
$d_v$ = 32
$d_k$ = 32
h = 16 |
| | Transformer-K2 | $d_{model}$ = 512
$d_v$ = 64
$d_k$ = 64
h = 8 |
| | Transformer-K3 | $d_{model}$ = 512
$d_v$ = 128
$d_k$ = 128
h = 4 |
| | Transformer-K4 | $d_{model}$ = 512
$d_v$ = 512
$d_k$ = 512
h = 1 |

Figure 12: Specifications of MLP and Transformer (Vaswani et al., 2017)

| Model | Feature name | Description |
|---|---|---|
| Hardware | mesh_x_ratio | The ratio of PE array and global buffer along x-axis |
| | mesh_y_ratio | The ratio of PE array and global buffer along y-axis |
| Software | input_buffer_usage | input data size / input (local) buffer size |
| | weight_buffer_usage | weight data size / input (local) buffer size |
| | output_buffer_usage | output data size / input (local) buffer size |
| | global_buffer_usage | all data size / global buffer size |
| | parallelism_ratio_x | used parallelism / available parallelism in the x-axis of global buffer |
| | parallelism_ratio_y | used parallelism / available parallelism in the y-axis of global buffer |

Figure 13: Extra features used by the hardware and software BO optimizers.

```
for n in [0:N]
  for k in [0:K]
    for r in [0:R]
      for s in [0:S]
        for p in [0:P]
          for q in [0:Q]
            for c in [0:C]
              outputs[n][k][q][p] += weights[k][c][s][r] *
                                     inputs[n][c][q+s][p+r]
```

Figure 14: Computing a 2D convolution with a seven-level nested loop.

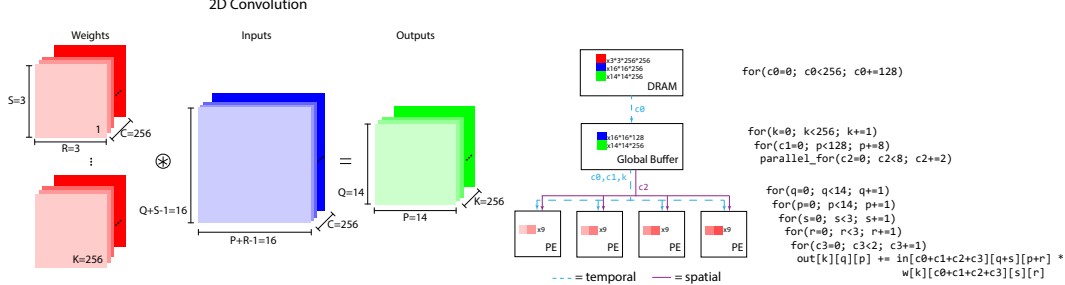

Figure 15: An architecture computing the CONV4 layer of ResNet.

| Index | Type | Range of Values |
|---|---|---|
| 1 | int | Factors of 256 |
| 2 | int | Factors of 256 |
| 3 | int | 0-220 (total local buffer size) |
| 4 | int | 0-220 (total local buffer size) |
| 5 | int | 0-220 (total local buffer size) |
| 6 | int | Factors of 168 |
| 7 | int | Factors of H1 |
| 8 | int | Factors of H2 |
| 9 | int | Factors of 16 |
| 10 | int | Factors of 16 |
| 11 | categorical | 0, 1 |
| 12 | categorical | 0, 1 |

Figure 16: An example vector of hardware parameters. Please refer to Figure 6 for more detailed descriptions.

| Index | Type | Range of Values |
|---|---|---|
| 1-2 | int | Factors of 3 |
| 3-4 | int | Factors of 3 |
| 5-6 | int | Factors of 28 |
| 7-9 | int | Factors of 28 |
| 10-12 | int | Factors of 128 |
| 13-17 | int | Factors of 128 |
| 18 | categorical | 0-1 |
| 19 | categorical | 0-5 |
| 20 | categorical | 0-1 |
| 21 | categorical | 0-1 |
| 22 | categorical | 0-23 |

Figure 17: An example vector of software parameters (with ResNet-K2). Please refer to Figure 8 for more detailed descriptions. In this example, parameters 1-17 correspond row-wise to S1-S6 respectively, parameters 18-20 correspond to S7, and parameters 21 and 22 correspond to S8 and S9 respectively.

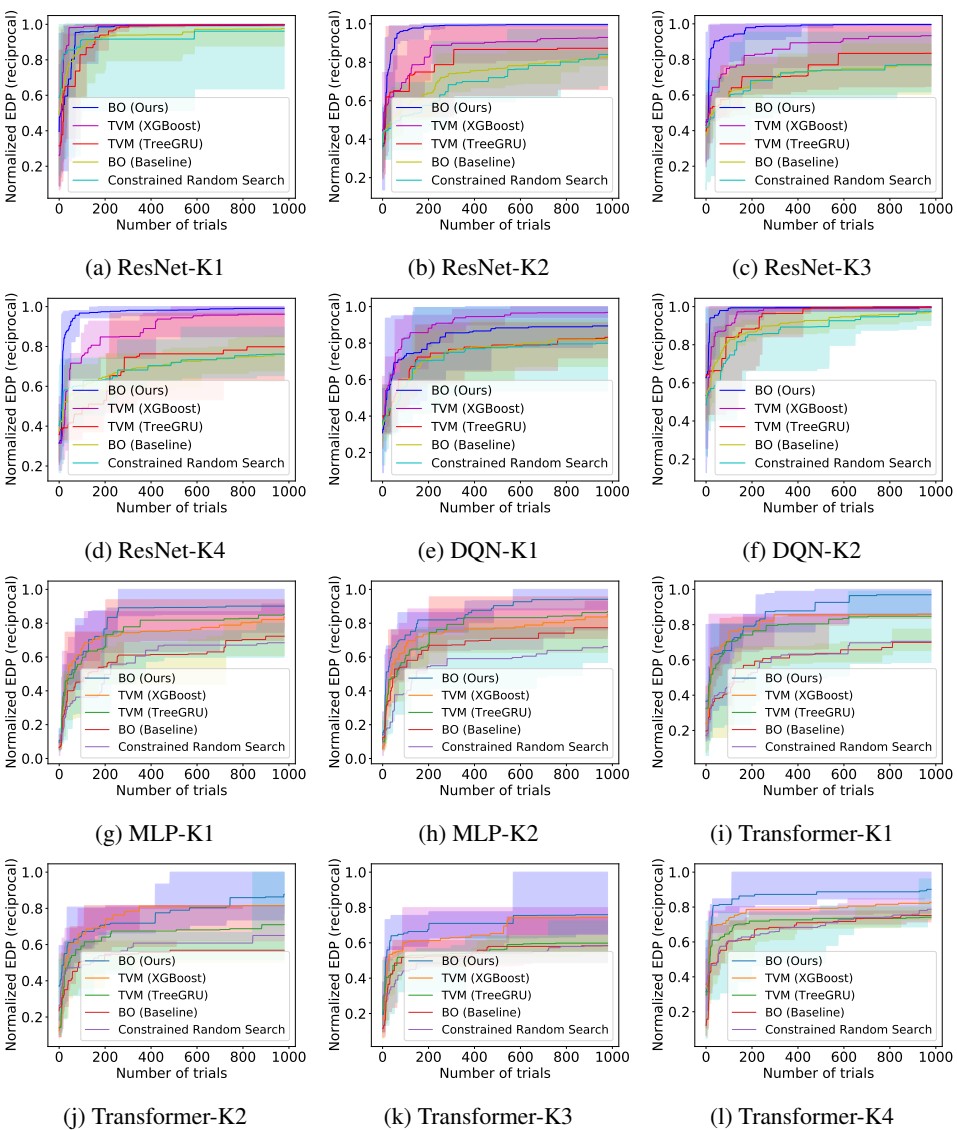

Figure 18: Software mapping optimization on ResNet, DQN, MLP, and Transformer. The Y-axis shows the reciprocal of energy-delay product (EDP) (normalized against the best EDP value). Higher is better.

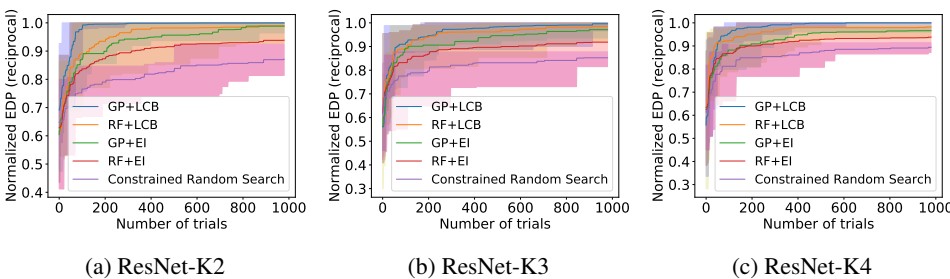

Figure 19: GP with different surrogate models and acquisition functions.

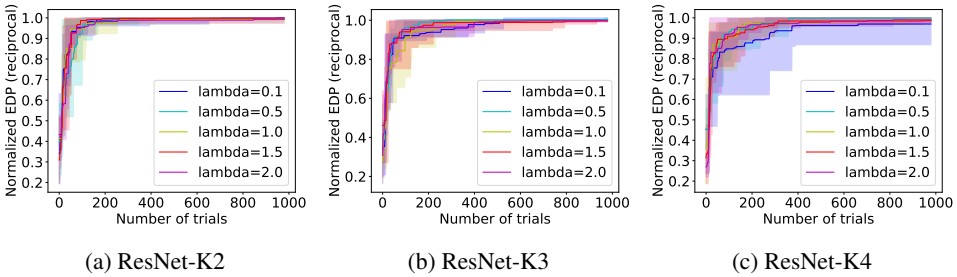

(a) ResNet-K2        (b) ResNet-K3        (c) ResNet-K4

Figure 20: LCB acquisition function with different lambda values.

