# OpenReview forum: "LEARNED HARDWARE/SOFTWARE CO-DESIGN OF NEURAL ACCELERATORS"
_ICLR.cc/2021/Conference — Reject_

### Official Review · AnonReviewer2 · 2020-10-26
**The paper needs to revise its claims**

**Rating:** 6
**Confidence:** 3

**Review:**

##########################################################################

Summary:

The paper presents a method for hardware and software co-design using Bayesian optimization method. It is based on previous work of Timeloop

This paper seems a just an application of Bayesian optimization on the Timeloop framework. It searches using set parameters that are tunable by using some optimization method. Nevertheless, the paper shows improvement upon state-of-the-art neural networks such as transformers.

##########################################################################

Reasons for score:

The paper shows improvement upon state-of-the-art neural networks such as transformers.

The paper being first system that co-optimizes both the hardware and software for DNN, which is dubious given the pletora of NAS and HAS papers.

##########################################################################

Pros:

- Demontrated improved energy delay design on different workloads

##########################################################################

Cons:

- The paper needs to revise its claim of being the  first system that co-optimizes both the hardware and software for DNN

- Evaluate using different machine learning optimization methods such as RL and Evolutionary algorithms

##########################################################################

Questions during rebuttal period:


Please address and clarify the cons above

---

> ### Author Response · Authors · 2020-11-16
> **Author response**
>
> Thanks for taking the time to review our work. We now address your comments below.
>
> **Q1**: Our claim of being the first system that co-optimizes both the hardware and software for DNN
>
> **A1**:  We will clarify this in the revision. Our goal is to optimize the system stack (i.e. hardware and compiler optimizations) for a given neural network, and we believe this is the first work aims to do so. Please refer to our response to reviewer 3 for details.
>
> **Q2**:  Evaluate using different machine learning optimization methods such as RL and Evolutionary algorithms
>
> **A2**:  The main hurdles for hardware/software co-design are sample efficiency and a highly constrained hardware search space with slow simulations. Provided these parameters, constrained bayesian optimization is a good fit for the problem. RL here is perhaps less ideal, the hardware domain is not data-rich and RL is not generally considered to be sample efficient. Evolutionary algorithms also are an interesting direction. We do not mean to suggest that bayesian optimization is the only approach, and hope that future work will continue developing optimization algorithms.

---

> ### Author Response · Authors · 2020-11-20
> **Comments are appreciated**
>
> Thank you again for taking the time to review our work. We'd like to answer any more questions that you may have.

---

### Official Review · AnonReviewer3 · 2020-10-28

**Rating:** 4
**Confidence:** 5

**Review:**

Review:
This paper proposes a Bayesian optimization framework for exploring the hardware configuration and compute mappings on generated hardware. Given a target NN model, it first generates hardware configuration candidates. Then for each hardware candidate, it generates corresponding mapping candidates of each NN layer. These candidates are generated with explicit constraints, i.e., it will randomly sample hardware/mapping until it obtains the desired number of feasible candidates. The evaluated EDP is fed into the Bayesian optimizer of mapping and hardware separately in sequential. The main contribution of this paper is a principled and systematic pipeline for co-searching the hardware parameters and mappings. Results are shown on the task of ResNet, DQN, MLP, and Transformer on Eyeriss architecture. It improves EDP by 16% to 40%.


Pros:

+ The solution framework for co-searching hardware parameters and software mappings is neat.

+ The results section is well structured.  It's nice to see the comparison between the proposed method and TVM; and the ablation study on whether to co-search hardware and software.


Concerns:

- The key concern about the paper is the lack of novelty. There are increasing works on hardware search for neural networks (see below), while this paper doesn't mention or compare to any of those.

- The related work lacks work on hardware search, such as
[1] Yuhong Li, Cong Hao, Xiaofan Zhang, Xinheng Liu, Yao Chen, Jinjun Xiong, Wen-mei Hwu, and Deming Chen. EDD: Efficient differentiable DNN architecture and implementation co-search for embedded AI solutions. arXiv preprint arXiv:2005.02563, 2020.
[2] Weiwen Jiang, Qiuwen Lou, Zheyu Yan, Lei Yang, Jingtong Hu, X Sharon Hu, and Yiyu Shi. Device-circuit-architecture co-exploration for computing-in-memory neural accelerators. IEEE Transactions on Computers, 2020a.
[3] Lei Yang, Zheyu Yan, Meng Li, Hyoukjun Kwon, Liangzhen Lai, Tushar Krishna, Vikas Chandra, Weiwen Jiang, and Yiyu Shi. Co-exploration of neural architectures and heterogeneous ASIC accelerator designs targeting multiple tasks. arXiv preprint arXiv:2002.04116, 2020.
[4] Yujun Lin, Driss Hafdi, Kuan Wang, Zhijian Liu, and Song Han. Neural-Hardware Architecture Search. NeurIPS Workshop, 2019.
[5] Mohamed S Abdelfattah, Łukasz Dudziak, Thomas Chau, Royson Lee, Hyeji Kim, and Nicholas D Lane. Best of both worlds: Automl codesign of a CNN and its hardware accelerator. arXiv preprint arXiv:2002.05022, 2020.

-  The results are not conclusively in favor of the proposed method. Why do authors adopt Bayesian optimization instead of evolution or reinforcement learning? How is Bayesian optimization reasonable for searching loop reordering? There is no discussion of the choice of optimization algorithm from the results.

- On the same note, the results would have been more complete if experiments can be done on multiple hardware architectures such as NVDLA, ShiDianNao, and so on.

- It would be better if the paper can elaborate on Section 2 in more detail. How does the proposed framework represent hardware and mapping? What is the range of each dimension? How large is the design space? What is the sample efficiency of the proposed method?

---

> ### Author Response · Authors · 2020-11-13
> **Put our work in the context of prior work**
>
> Thank you for the detailed reviews. Before we respond to individual concerns, we wanted to address the shared concern of our claims vs. prior work.
>
> We apologize for not clearly placing our work in context. As the reviewers have pointed out, there are three major categories of prior work when considering automatic optimization of neural models.
>
> The first is neural architecture search (NAS), which considers the optimal neural network topology to use for each problem, often agnostic to the underlying hardware or software. We viewed NAS as disjoint to our work originally, but will add citations and a discussion to clarify.
>
> The second is software compiler optimizations (SCO) (like TVM [6]), which considers the optimal software mappings, such as loop permutations and tiling, to apply to a given neural network.
>
> The third is hardware architecture search (HAS), which governs the optimal hardware to use provided a particular neural architecture and software optimization. Optimization parameters often include buffer sizes and PE layout.
>
> Originally, each of these problems were studied independently, but as you’ve pointed out, recent research has started to optimize pairs of categories jointly. For example, EDD [1] co-optimizes NAS and SCO on embedded systems, but doesn’t consider HAS. The SCO parameters they consider are equivalent to our software evaluation.  [2,3,4,5] all co-optimize NAS and HAS and couple SCO with HAS. They either fix the compiler optimizations [2, 4, 5] or tie the compiler optimizations (e.g. dataflows) to the hardware search [3]. This unified space of NASAIC [3] simplifies the search problem, but the compiler optimizations cannot be customized to different workloads once the hardware is synthesized.
>
> In our work, we propose jointly optimizing the system stack (HAS + SCO) for popular variants of neural networks.  Timeloop [7] shows that it is necessary to optimize the software mapping (SCO) for the hardware architecture to fully exploit the benefits of the hardware. Because hardware is much more costly to evaluate than compiler optimizations, we propose a novel nested search to decouple SCO from HAS, to 1) minimize expensive hardware generation, and 2) optimize hardware for the entire model but still optimize software mappings for each layer.
>
> We will clarify the novelty claim to reflect this discussion.
>
> [6] Chen, Tianqi, et al. "TVM: An automated end-to-end optimizing compiler for deep learning." 13th {USENIX} Symposium on Operating Systems Design and Implementation (OSDI) 2018. “Learning to optimize tensor programs." In Advances in Neural Information Processing Systems (NeurIPS), 2018
> [7] Parashar, Angshuman, et al. "Timeloop: A systematic approach to dnn accelerator evaluation." International symposium on performance analysis of systems and software (ISPASS). IEEE, 2019.

---

> ### Author Response · Authors · 2020-11-16
> **Responses to other comments**
>
> Please see our more detailed responses to the other comments.
>
> **Q1**: How is BO reasonable for searching loop reordering?
>
> **A1**: We represent different reordering options as categorical parameters, and empirically we find our GP does not suffer from searching in this discrete space.
>
> **Q2**: Evaluate on multiple architectures
>
> **A2**: We are working on evaluating more counterparts, and we will post our results once it’s done.
>
> **Q3**: How does the proposed framework represent hardware and mapping? What is the range of each dimension? How large is the design space? What is the sample efficiency of the proposed method?
>
> **A3**: We show the representation in Figure 6 and 8 in Appendix. The actual range of dimensions and scale of the design space depends on the hardware budget (e.g. 168 PEs and 220 buffer entries) and neural networks (see Figure 11 and 12). Our experiments show effective designs can be found using 50 hardware trials with each taking 250 software trials.

---

> > ### Author Response · Authors · 2020-11-25
> > **More results**
> >
> > During the rebuttal period we find that our approach improves EDP by 24% over Diannao on DQN. We are running our optimizer on other workloads with other search algorithms and will include results in the next version of the paper, but they have not finished in time for the discussion period.

---

> ### Author Response · Authors · 2020-11-20
> **More questions are welcomed**
>
> We'd like to thank you again for your comments, especially the one that asks us to put our work in the context. We hope that we've addressed the concerns, and we’d be happy to answer any questions that you may have.

---

### Official Review · AnonReviewer1 · 2020-10-28
**Interesting paper but needs more clarification**

**Rating:** 5
**Confidence:** 2

**Review:**

This paper presents a method to optimize the co-design of code mapping and hardware configuration for neural accelerators. The method is based on a two-level Bayesian optimization (BO), with each BO optimizing the code mapping and hardware configuration, respectively. The design space of both code mapping and hardware configuration contains many complex constraints, some of which remains unknown until running expensive simulation. The obvious constraints are handled using rejection sampling and the unknown constraints are modeled by a Baysian classifier. This method is validated by experiments to optimize several network layers (ResNet, DQN, MLP, transformer) and achieves better performance than baselines.

Strength of the paper: this paper aims at solving a meaningful problem: SW/HW co-design of code mapping and neural accelerators. It adopts a simple optimization method (BO), and according to the experiments, it achieves good performance gain compared with manual design and other baselines methods.

What is unclear to me: I am not familiar with the literature around code-mapping and hardware configuration co-design, so I don't have enough context to judge if the proposed method is novel, compared with previous methods.

Weaknesses of the paper:
1/ this paper spends a lot of texts explaining the basics of BO, but what's missing to me is the representation of software design-points and hardware configurations. The usage of BO is based on the assumption that we have a good representation of the design space where the distance of each design point is clearly defined. For example, for two nested for-loop with different orders, how they are represented and what's the distance between the two representation? Is the representation canonical? What's the meaning of distance in the representation space? Without understanding these questions, it is difficult for me to see whether BO is the right optimization method for this.
2/ In the experiment section, the method is only used to optimize one layer of a network, however, it is not clear how general this is as people usually care about the overall performance of a network or even a wide variety of networks. The experiments did not show if the proposed method is effective or general enough to optimize the overall performance.

I hope authors can address the concerns.

---

> ### Author Response · Authors · 2020-11-16
> **Author response**
>
> Thanks for the great comments. Please see our responses below.
>
> **Q1**: Representation and distance in the search space
>
> **A1**: This is an extremely good question and key to allowing the model to effectively optimize over the design space. The representation that we used is not the only one that could but used, however it is the most effective that we’ve found so far. We have tried a few other approaches to defining distance in this space. The challenge is that our constraints are highly nonlinear, which makes representations difficult. Initially, we applied log transformations to the tiling factor to transform the nonlinear constraints to linear. The log space works fine for numbers like 64 whose factors are all a power of a number, but fails for arbitrary numbers like 56. Taking a deeper look, we found that our GP suffers greatly from the transformed space where the distance is skewed. Our current representation sticks to the original values of the parameters, and we further enhance the GP with features that reveal high-level semantics, such as buffer and compute usage. To clarify the representation that we’ve used, we added example vectors of hardware/software parameters to the appendix (Figures 16 & 17 respectively).
>
> **Q2**: Layer-wise and model-wise optimization
>
> **A2**: As software optimization is more flexible, we optimize the software for each layer but optimize the hardware for each model. Our evaluation shows model-level improvements. We will clarify this.

---

> > ### Comment · AnonReviewer1 · 2020-11-23
> > **Response to author response**
> >
> > > A1
> > Thanks to the authors for your response and for providing more information. I believe that the representation of the search space should be regarded as a central piece since it is a determining factor for the effectiveness of the optimization algorithm. However, I don't think it is sufficiently discussed in the current paper. I think a convincing discussion (through rigorous analysis or extensive experiments) of the representation and the connection to the optimization algorithm is needed to present a strong case for the proposed method.
> >
> > > A2
> > SW layers are more flexible but only to certain degrees. For example, for a CNN used in image classification, the characteristics of an early layer vs. a late layer are very different. It is infeasible to assume that CNN layers can be arbitrarily adapted to fit hardware optimization. As a result, to show the method applies to the overall model, it is important to showcase the proposed method can optimize a wide range of layer configurations and show consistent improvement, and also show overall improvement on the model level.

---

> > > ### Author Response · Authors · 2020-11-25
> > > **Response**
> > >
> > > Thanks to the reviewer for your response. We now address the comments below.
> > > A1: This is a good question and an open research topic. However, our representation does provide state-of-the-art results and this work is the first to address the problem, particularly with BayesOpt. We added a paragraph in the paper discussing the representation, and we hope that future research can continue to improve automatic hardware/software codesign and provide new representations"
> > >
> > > A2: With the objective being the EDP for the entire model, the hardware is optimized to be effective for all layers, minimizing total EDP for a forward pass. Our objective is consistent with state-of-the-art manual designs, such as Eyeriss and Diannao, which adapt all software layers to a single piece of hardware.  We can provide layer-by-layer results in the next version, but they're generally similar to full model performance.

---

> ### Author Response · Authors · 2020-11-20
> **Happy to answer more questions**
>
> We'd like to thank you again for taking the time to review our work. We hope we've addressed your concerns, are there any additional questions that we could answer?

---

### Official Review · AnonReviewer4 · 2020-11-01
**This manuscript is interesting, and the work is promising. However, the formal representation of hardware and software lack important parameters. Moreover, the effectiveness of the proposed approached is evaluated against one previous work on a single metric.**

**Rating:** 7
**Confidence:** 4

**Review:**

In this paper, the authors propose to co-optimize the software and hardware for DNN executions to maximize the energy-delay product. There are three main contributions: 1) propose a formal representation of software and hardware that facilitate the search process; 2) propose a Bayesian optimization framework; 3) propose to search the hardware and software space separately and the search process is optimized by leveraging the Gaussian process model. Below are some questions and concerns:

1:  the representation of software basically considers loop ordering, loop tiling, and computational parallelism, while the hardware representation is focused on available hardware resource and dataflow configuration. However, the biggest bottleneck of DNN processing actually comes from the main memory or large buffers. Compared with computing itself, data access normally consumes 2 orders of magnitude more energy and latency. Could the authors explain if they consider data access during their search process and how they famulated in the representation?

2: the search goal of this work is to minimize EDP. I wonder is it flexible for this framework to change this goal to fit different designs? For instance, is it possible to search a software/hardware co-designed model for extremely low-power applications? And how can the proposed framework adopt for different search goals?

3: the evaluation is only performed with previous work Eyeriss. Given the abundant accelerator with various design goals, more evaluations against more counterparts are needed.

4: for the comparison results against Eyeriss, the authors show that their results outperform Eyeriss in EDP. But I am also curious about the detailed energy and performance comparison.

---

> ### Author Response · Authors · 2020-11-16
> **Author response**
>
> Thank you for taking the time to review our paper. We now address your comments below.
>
> **Q1**: Consider data access in search process
>
> **A1**: Timeloop associates high energy and latency costs with data accesses, and because our objective is EDP, we implicitly optimize data accesses.  In particular, the hardware BO optimizes the buffer sizes and allocations at each level of storage, and the software BO optimizes the use of buffers through loop tiling and reordering.  Expanding to larger memory structures (like HBM) could also be considered.
>
> **Q2**: Change the optimization objective
>
> **A2**: Optimizing for a different objective is feasible. To achieve high sample efficiency, we might need to modify the feature design and the GP model accordingly. For example, if we target applications with a strict energy consumption limit, we can treat them as output constraints and use extra GPs to incorporate the constraint in the acquisition function (e.g. constrained EI).
>
> **Q3**: More (detailed) evaluations
>
> **A3**: We are working on evaluating more counterparts, and we will post our results once it’s done.

---

> > ### Author Response · Authors · 2020-11-25
> > **More results**
> >
> > We find that our approach improves EDP by 24% over Diannao on DQN. More results (other workloads and search algorithms) have not finished in time for the discussion period, but we're working on them and will include results in the next version of the paper.

---

> ### Author Response · Authors · 2020-11-20
> **Thanks for the comments**
>
> We'd like to thank you again for the great comments. We hope that we've addressed your concerns and we'd be happy to discuss in more details if needed.

---

### Decision · Program_Chairs · 2021-01-07
**Final Decision**

**Decision:**

Reject

**Comment:**

This paper considers the problem of hardware and software co-design for neural accelerators. Specifically, it looks at hardware and the software compiler that maps DNN to hardware. It employs Bayesian Optimization (BO) to perform joint search over hardware and software design parameters in an alternating manner. To handle black-box constraints that cannot be evaluated without performing simulations, the method uses constrained BO algorithms.

The paper talks about two technical challenges:
1) Black-box constraints. There is a lot of literature on constrained BO.
2) Semi-discrete design variables. The paper didn't propose any generic solution. There are some recent papers to handle mixed variables that may be useful.
https://arxiv.org/abs/1907.01329
https://arxiv.org/abs/1906.08878

BO methodology is justified. There is recent work on hardware and software co-design for neural accelerators and should be taken into account for both qualitative and quantitative comparison.

Overall, my assessment is that the paper in its current form lacks technical novelty for acceptance.